# Malaria and Hantavirus Pulmonary Syndrome in Gold Mining in the Amazon Region, Brazil

**DOI:** 10.3390/ijerph16101852

**Published:** 2019-05-25

**Authors:** Ana Cláudia Pereira Terças-Trettel, Elaine Cristina de Oliveira, Cor Jesus Fernandes Fontes, Alba Valéria Gomes de Melo, Renata Carvalho de Oliveira, Alexandro Guterres, Jorlan Fernandes, Raphael Gomes da Silva, Marina Atanaka, Mariano Martinez Espinosa, Elba Regina Sampaio de Lemos

**Affiliations:** 1Nursing Department, Mato Grosso State University Campus Tangará da Serra, Tangara da Serra (MT) 78300-000, Brazil; 2Department, Mato Grosso Federal University, Cuiaba (MT) 78000-000, Brazil; corfontes@gmail.com (C.J.F.F.); marina.atanaka@gmail.com (M.A.); marianomphd@gmail.com (M.M.E.); 3Epidemiology Department, Health Secretary of State of Mato Grosso, Cuiaba (MT) 78000-000, Brazil; elainecristina.mt@gmail.com (E.C.d.O.); albagomes@gmail.com (A.V.G.d.M.); 4Hantaviruses and Rickettsiosis Laboratory, Oswaldo Cruz—FIOCRUZ Institute, Rio de Janeiro (RJ) 21000-000, Brazil; reoliveira@ioc.fiocruz.br (R.C.d.O.); guterres@ioc.fiocruz.br (A.G.); jorlan@ioc.fiocruz.br (J.F.); raphael@ioc.fiocruz.br (R.G.d.S.); elemos@ioc.fiocruz.br (E.R.S.d.L.)

**Keywords:** malaria, hantavirus pulmonary syndrome, infectious diseases epidemiology, differential diagnosis

## Abstract

People living in mining regions are exposed to numerous biological agents by several specific types of transmission mechanisms. This study is designed to describe fatal hantavirus pulmonary syndrome (HPS) cases confirmed by serology and molecular analysis, where a seroprevalence survey was conducted in the gold mining regions of the state of Mato Grosso, in the official Amazon region, Brazil. Two fatal cases of HPS were confirmed in a mining area in the Legal Amazon, where malaria is one of the most important public health problems. A molecular analysis detected the presence of the genome of the Castelo dos Sonhos virus. Out of the 112 blood samples analyzed, five were positive for *Plasmodium* infection (four *P. falciparum* and one *P. vivax*), and four were seropositive for hantavirus, showing a seroprevalence of 3.57%. One of the four miners who was seroreactive for hantavirus concomitantly had *P. falciparum* infection, which was confirmed by thick blood smear. This manuscript highlights the importance of considering hantavirus pulmonary syndrome as a diagnostic possibility in febrile infection associated with pulmonary manifestations in mining areas where malaria cases are often identified.

## 1. Introduction

People living in mining areas are subject to adverse conditions that are often dangerous, painful and unhealthy. The miners’ job is one of the most precarious and intense existing occupations [1]. This vulnerability exposes them to different risks, such as hearing impairment, disc herniation, repetitive stress injury, skin and urinary problems, respiratory system diseases, febrile illnesses, malaria, and long-term cancer [2,3].

The occurrence of malaria cases in gold mining regions has been reported in the countries of Africa, Asia [4,5,6], and South America (Brazil, Colombia, Guyana, Peru, Suriname and Venezuela) [7,8,9,10,11,12]. These occurrences are related to the profound environmental changes that mining activities cause, disrupting the existing ecological balance in areas that were previously preserved.

In Brazil, most malaria cases come from rural areas associated with gold mining [12,13]. The gold mining areas of northern Mato Grosso greatly contribute to the numbers of malaria cases, and these populations are exposed to numerous vulnerabilities, such as poor housing quality due to physical proximity to the workplaces, poor access to health services, and a modified environment that favors the presence of vectors and wild animals [14,15,16].

Because of both close contact with a wild environment that has undergone extensive modification and the unspecific health problems identified in this population, it is likely that emerging diseases are occurring and not diagnosed, possibly because of the difficulty of access to health services.

Hantavirus pulmonary syndrome (HPS), also known as hantavirus cardiopulmonary syndrome (HCPS), is an emerging, acute, severe, and highly lethal disease caused by several genotypes of hantavirus and transmitted by wild rodents in the American continent. Since the first evidence of hantavirus circulation in the Brazilian official Amazon region in 1991, 124 confirmed cases have been identified, associated with the genotypes of the Anajatuba, Castelo dos Sonhos and Rio Mamore viruses. Cases of HPS have been reported in the legal Amazon region as a consequence of, among other factors, major changes in the environment and close contact between humans and wild animals, mainly rodents [17,18,19,20,21,22].

In South America, the hantavirus rodent reservoirs that cause HPS belong to the subfamily Sigmodontinae, and in Brazil, where there is a wide diversity of wild rodents distributed in different biomes, six hantavirus genotypes associated with SPH have been described: Juquitiba (also identified as Araucaria), Araraquara, Castelo dos Sonhos, Laguna Negra-like, Anajatuba, and Rio Mamoré [21,22]. In the state of Mato Grosso, where the study was conducted, two genotypes of hantavirus are associated with HPS: (i) Castelo dos Sonhos, responsible for the miners’ cases, maintained in the rodent reservoir *Oligoryzomys utiairitensis* previously identified in this region [23,24] and (ii) the Laguna Negra, also described in the regions close to the mining of Mato Grosso, for which the reservoir is the wild rodent *Calomys callidus* [25].

It is noteworthy that the characteristic signs and symptoms of the prodromal phase of HPS are nonspecific, including fever, myalgia, malaise, headache, chills, nausea, and vomiting; thus, the differential diagnosis method is essential to distinguish HPS from other diseases that have similar initial characteristics [26,27,28,29,30]. In fact, although infrequent, the appearance of respiratory complications related to malaria can simulate HPS in the cardiopulmonary stage when the patient has dry cough, accompanied by tachycardia, dyspnea, and hypoxemia, followed by rapid progression to pulmonary edema, hypotension, and circulatory collapse [29,31,32,33,34,35].

In this context, the objective of this study is to describe the occurrence of hantavirus in the gold-mining regions of Mato Grosso state, Brazil, emphasizing the importance of differential diagnosis.

## 2. Cases

### 2.1. Case Presentation

This study reports two cases of HPS in the gold mining region of the União do Norte district, Peixoto de Azevedo in northern Mato Grosso (Figure 1), and a study of seroprevalence in the population of a region located to the west of the occurrence of confirmed cases.

### 2.2. Case Report

In this study, HPS was confirmed, by serology and molecular analysis, in two miners who worked in the far northern part of Mato Grosso, in the União do Norte district, city of Peixoto de Azevedo, a malaria transmission area. The epidemiological investigation of HPS cases revealed that the patients were exposed to aerosolized dust while cleaning the accommodations where they lived and worked one week before the occurrence of the first symptoms. In addition, they also had a history of consuming fruits collected from the ground in rodent-infested areas. The lack of previous notifications about HPS cases among miners and the frequent occurrence of malaria and arboviruses in the population of the mining areas were determinants in the fatal evolution of the two clinical cases.

Patient 1 (gold mining region of the União do Norte district, Peixoto de Azevedo in northern Mato Grosso): 

In June 2015, a 37-year-old male patient presented to a public hospital with a history of fever, headache, and myalgia; he was treated and released with suspected dengue. Four days later, the patient presented with dyspnea, acute respiratory failure, blurred vision, and chest pain, and was referred to the intensive care unit (ICU), where treatment included the use of antibiotics and a mechanical respirator. His nonspecific tests showed increased urea and creatinine (127.40 mg/dL and 2.42 mg/dL, respectively), thrombocytopenia (58,000/mm^3^), and leukocytosis (20,040/mm^3^). A chest radiograph confirmed a pulmonary diffuse interstitial infiltrate (Figure 2). On the sixth day of the disease, with suspected HPS, a blood sample was collected, of which the serological analysis confirmed the presence of anti-hantavirus IgM antibodies with negative IgG [36]. Hantavirus genome was detected in a blood sample using reverse transcription polymerase chain reaction (RT-PCR), and the genotype identified was Castelo dos Sonhos virus [37]. Despite the measures imposed in the intensive care unit, the patient progressed to death three weeks after the onset of illness. The patient was a machine operator in a gold mine and lived in Sinop, a municipality in the Legal Amazon region in Mato Grosso state.

Patient 2 (gold mining region of the União do Norte district, Peixoto de Azevedo in northern Mato Grosso): 

A 47-year-old man with fever, headache, myalgia, chest pain, dry cough, dizziness, asthenia, dyspnea, acute respiratory failure, and back pain was admitted to the same public hospital as Patient 1 in June 2015. The laboratory test revealed hemoconcentration (47.1%), thrombocytopenia (37,000/mm^3^), leukocytosis (22,180/mm^3^), increased urea and creatinine (72.56 mg/dL and 1.87 mg/dL, respectively), aspartate aminotransferase (102.4 IU), and alanine aminotransferase (57.14 IU). Chest X-ray was not performed. The analysis of the serum sample collected on the seventh day of the disease showed the presence of anti-hantavirus IgM antibodies, with negative IgG, but detected hantavirus Castelo dos Sonhos by RT-PCR. Although a therapeutic strategy based on antibiotics associated with hemodynamic and respiratory support was followed, the patient died nine days after the onset of the illness.

After the first occurrence of HPS in miners, a study was carried out to estimate the prevalence of anti-hantavirus antibodies in 112 samples previously collected from a population living in a mining area in the Três Fronteiras district in the city of Colniza, Mato Grosso (Figure 1). These serum samples, which were stored in the Malaria Biorepository of the University Hospital Júlio Muller following a malaria survey conducted in 2012, were used due to the physiogeographical and population similarities between this gold mining area and the area where the two fatal HPS cases were identified. These two mining areas, even if geographically distant, are comparable because they are located in the Amazon biome, have the same environmental modifications from the garimpo and the populations have similar income, housing and access to health.

The data were collected in July 2012 from 112 Igarapé Grande gold mining and São Francisco gold mining, municipality of Colniza, this number represents all inhabitants of the mining areas mentioned above. This data included collection of blood samples using the finger prick and thick smear technique, completion of the SIVEP-Malaria notification form, and completion of an interview to obtain demographic and socioeconomic information and information about exposure to malaria transmission.

The serum samples from human cases of HPS and cohort were tested by anti-hantavirus IgG and IgM antibodies screening, using the recombinant N protein of Araraquara virus, provided by the University of São Paulo/Ribeirão Preto [38], following the protocols of enzyme immunoassays ELISA. This antigen is representative for all genotypes isolated in Brazil.

The malaria incidence in the population of miners in Colniza in 2012 was 4.46% (4 cases of *Plasmodium falciparum* and one of *P. vivax*), whereas the hantavirus seroprevalence was 3.57%, with four reactive IgG samples, all negative for IgM antibodies. One of the four hantavirus seropositive patients, in addition to mentioning an unspecific fever history on the day of data collection, also presented positive results in a thick blood smear for malaria with the identification of *P. falciparum*.

For demographic data, among the 112 study participants, 56.25% were men. Nevertheless, when evaluating the four hantavirus seroreactive patients, three were women. The age of the study population ranged from six months to 65 years, with an average of 29 years. It is noteworthy that one of the 48 women was pregnant. The predominant color was *pardo* in 68.8% of the general population and 50% of the seroreactive population, while 41.1% of the study population were married and another 41.1% single, see Table 1.

Regarding educational attainment, Table 1 shows that all seropositive patients and 75% of the total attended school. Vegetal exploration (18.75%), which include professional activities that involve explorations of the environment, and housewife (12.5%) were the most common occupations, but 64.4% of the respondents mentioned other types of employment related to mining activity. The most common housing type was wooden houses, in addition to seven houses built from canvas. Eight interviewees reported information collected on clinical aspects, and one hantavirus seroreactive individual reported fever, headache, and body pain, see Table 2.

## 3. Discussion

The clinical manifestations of acute febrile diseases are nonspecific and may hinder diagnosis and clinical management, especially in relation to infectious diseases such as HPS and malaria that can often coexist in endemic areas, as well as dengue, chikungunya, zika, and leptospirosis, among others. Both malaria and HPS, in the prodromal phase, have fever, myalgia, malaise, headache, abdominal discomfort, chills, nausea, and vomiting as clinical manifestations, which reinforces the need to consider differential diagnosis in areas with eco-epidemiological conditions suitable for vector and rodent transmissions as in the mining areas of this study [26,29,39].

In HPS, hemoconcentration, leukocytosis, atypical lymphocytes and thrombocytopenia, and elevated serum levels of liver enzymes are generally observed [27,40,41,42,43]; laboratory findings are similar to those observed in malaria and dengue [44,45,46,47,48]. In relation to chest radiographs in HPS, frequently observed findings include bilateral diffuse interstitial infiltrates, pulmonary edema, and pleural effusion during the cardiorespiratory phase [29,49]. Respiratory involvements in malaria are rare, but well-documented; they can be confused, however, with the severe phase of HPS [31,32,34,35]. In this context, the occurrence of HPS must be considered in patients with acute febrile illness associated with respiratory failure in areas where environmental changes can facilitate hantaviral transmission.

According to the National Department of Mineral Production, there are 185,832 workers in mineral extraction and 8368 mining regions officially registered in Brazil, although these numbers may be underestimates due to the informality of this professional activity [44]. Thousands of people living in the mining areas of the Brazilian Amazon are subject to numerous risks that can influence their health condition and quality of life.

In our study of seroprevalence, the most frequent housing type was wooden houses, but there were also canvas houses, demonstrating the social fragility to which the miners are exposed and the resulting risk of contact with animals that transmit the different diseases described herein. These existing housing conditions facilitate the transmission of causative agents of both malaria and HPS since the cracks in the walls of these residences facilitate the entry of the malaria mosquito and wild rodents searching for food and shelter [15,17,45]. Environmental changes resulting from mining activities have continuously influenced the habitat and species composition of wild rodents with consequent impacts on the health conditions of the people living there. These influences are already observable in the high number of cases of malaria in these regions [12,13] and can also be observed in the transmission of hantavirus, since the predators and competitors of rodents are usually exterminated in the habitats altered by human activities, an event that favors, in due course, the increased density of rodents and the establishment of new species. Thus, environmental changes caused in nature by the exploitation of mining activities favor the emergence of new diseases such as HPS [19,46,47].

Studies conducted to clarify doubts in the diagnosis of febrile disease of unknown etiology point to the need for immediate laboratory confirmation during outbreak/epidemic situations, as the many clinical manifestations may not characterize the diseases [48,49]. Serologic evidence of hantavirus has been identified in patients with clinical suspicion of dengue, malaria, influenza, chikungunya, rickettsial infections, leptospirosis, and HIV [50,51,52,53,54,55,56,57,58,59]. Thus, the results of this study in which the initial diagnosis was malaria and dengue, reinforce the need for an alert to health professionals working in endemic areas of malaria regarding the importance to investigate the hantavirus infection, as here performed with the population from Colniza.

One of the 112 participants of the malaria survey, who was *P. falciparum* positive, showed immunoserological evidence of hantavirus infection. Concomitant hantavirus infection with other agents, such as *Leptospira, Mammarenavirus, Dengue virus, hepatitis B virus*, and *Mycobacterium,* have been reported rarely as pulmonary malaria and HPS, the two important and fatal infectious diseases that are clinically indistinguishable and may exist in endemic areas; this coinfection, however, is rarely investigated. Thus, although the possible false reactivity should be considered in interpreting this result, the identification of two fatal HPS cases and serological evidence of infection in another three malaria negative miners (one of them with history of fever, headache, and body pain), reinforce the possibility of concomitant malaria–hantavirus infection in patients living in this Brazilian malaria-endemic area. A positive IgG assay with IgM negative serology indicates past infection because IgM antibodies appear early after infection and can remain, on average, for 30 days after the onset of infection [60,61,62,63]. However, it is important to recognize that at the onset of hantavirus infection, IgG antibodies are also detectable concomitantly with IgM antibodies. In this context, there may be a possibility of dual infection of malaria and HPS in the miner from Colniza in 2012, given that he had unspecific febrile illness.

Lastly, although there are no other studies that describe the presence of HPS in mines, it is possible to compare this finding with hantavirus seroprevalence studies conducted in populations from different regions of Brazil, where rates have ranged from 0.52% to 13.2% [64,65,66,67,68,69,70]. It is noteworthy that in cluster situations as described by Terças et al. [71] in the indigenous community in far northern Mato Grosso, the seroprevalence can be high and reach 51.1%.

In Brazil, the Amazon region accounts for 99% of malaria cases, and the incidence of malaria in gold mining regions is proportional to deforestation, being present in 39 cities of this biome [72,73]. Although the incidence rates may be even higher (31.3 to 94%), there has been a reduction in recent years in response to control programs implemented by public health services, which can be seen in the gold mining region in Colniza, where there was a seroprevalence of 4.5% [14,15,16,44,74,75].

## 4. Conclusions

This study presents two fatal cases of HPS confirmed by serological and molecular analysis in miners in the União do Norte district of Peixoto de Azevedo, Mato Grosso and shows that the genotype Castelo dos Sonhos virus infected the patients. Malaria and HPS may have the same level of seroprevalence in the study area of Mato Grosso.

Considering the absence of previous serological surveys for hantavirus in vulnerable populations residing in the mining area, the detection of hantavirus antibodies in the mining region in the city of Colniza, Mato Grosso reinforces the need for vigilance in preventing this rodent-borne disease in the Brazilian gold mining regions.

The confirmation of hantavirus cases in the mining region of far northern Mato Grosso prompts reflection in relation to emerging diseases in vulnerable and more susceptible populations due to their constantly changing environments. In this scenario, the investigation of other infectious agents, particularly zoonotic pathogens such as hantavirus, should be encouraged by health services because the correct diagnosis will direct proper and effective assistance.

Finally, this study highlights the importance of including HPS in a differential diagnosis and performing hantavirus screening among patients with febrile illness living in malaria-endemic areas, particularly in individuals with pulmonary failure.

## Figures and Tables

**Figure 1 ijerph-16-01852-f001:**
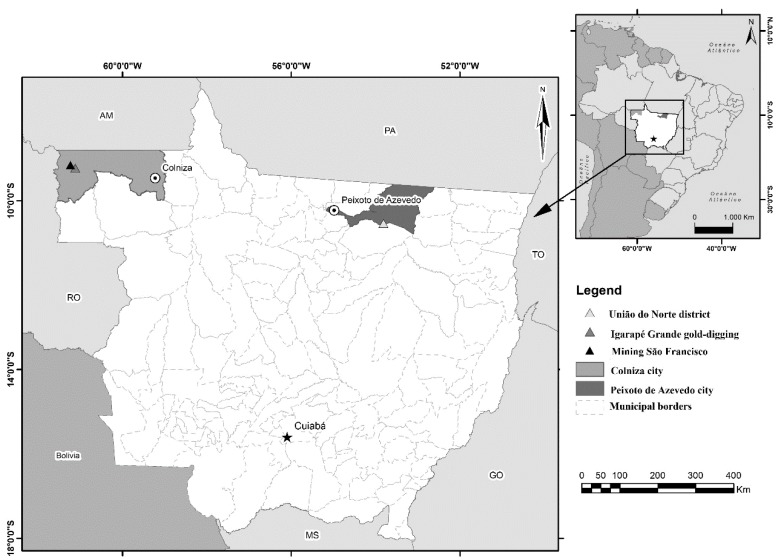
The geographic location of the study areas, gold mining regions of Mato Grosso, Brazil 2016.

**Figure 2 ijerph-16-01852-f002:**
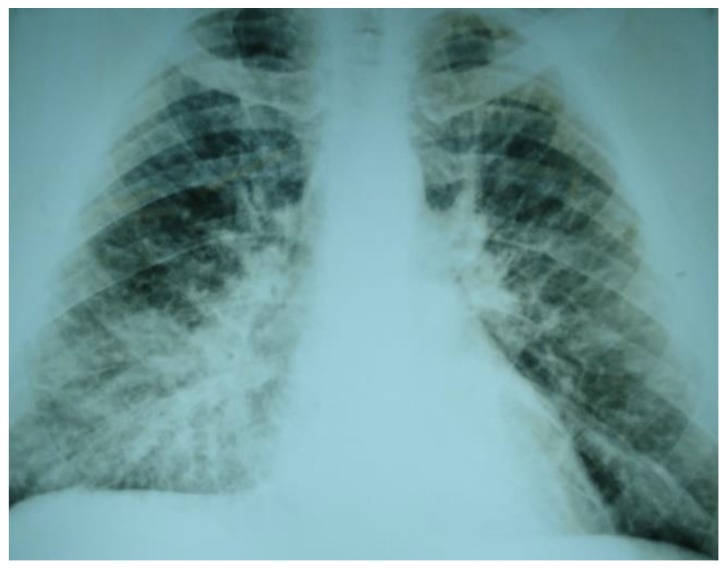
An X-ray of Patient 1 at the time of admission to the intensive care unit (ICU), Mato Grosso, Brazil, 2015

**Table 1 ijerph-16-01852-t001:** The socio-demographic characteristics of 112 residents of Três Fronterias, state of Mato Grosso, Brazil, 2012.

Independent Variables		Anti-hantavirus IgG + Patients		Total	
		*N*	%	*N*	%
Gender	Male	1	25	63	56.25
	Female	3	75	49	43.75
	Total	4	100	112	100
Educational Attainment	No education	-	-	13	11.6
	Elementary School	4	100	84	75
	High School	-	-	14	12.5
	Higher Education	-	-	1	0.9
Race Color	White	1	25	20	17.8
	Black	1	25	15	13.4
	Pardo	2	50	77	68.8
Occupation	Livestock activity	-	-	2	1.7
	Agricultural activity	-	-	3	2.7
	Housewife	-	-	14	12.5
	Vegetal exploration	1	25	21	18.75
	Other mining activity	3	75	73	64.4
Marital Status	Single	1	25	46	41.1
	Married	2	50	46	41.1
	Divorced	-	-	4	3.5
	Widower	-	-	1	0.9
	Consensual Union	1	25	15	13.4
Housing Type	Wood	4	100	104	92.8
	Canvas	-	-	7	6.3
	Other	-	-	1	0.9

**Table 2 ijerph-16-01852-t002:** The clinical characteristics of 112 residents of Três Fronterias, state of Mato Grosso, Brazil, 2012.

Independent Variables		Anti-hantavirus IgG + Patients		Total	
		N	%	N	%
Signs and symptoms during data collection	Yes	1	25	8	7.2
	No	3	75	104	92.8
	Total	4	100	112	100
Thick Blood Smear	Negative	3	75	107	95.5
	Positive for *P. falciparum*	1	25	4	3.6
	Positive for *P. falciparum* + Falciparum gametocity (FG)	-	-	-	-
	Positive for *P. vivax*	-	-	1	0.9
Reported co-morbidity	Yes	1	25	10	8.9
	No	3	75	102	91.1

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
