# Peer review of "Malaria and Hantavirus Pulmonary Syndrome in Gold Mining in the Amazon Region, Brazil"

_ijerph, 2019, doi:10.3390/ijerph16101852_

Round 1

Reviewer 1 Report

In this study, the authors describe the occurrence of HPS and malaria in the particular context of gold mining.This stud is of interest as it is highlighting the difficulty to diagnose the etiology of febrile syndrome in  tropical area and particularly in poor settings.The results of this study reinforce the the idea that that it is necessary to carry our a diagnosis of malaria before administering anti malarial treatment.

The study design is very clear and the results are well presented

However, the manuscript could be improved. In the introduction section the authors gave in a concise manner the prevalence of malaria and HPS in brasil by citing references, it would be nice to let know how these diseases, particularly HPS is occurring in the context of south America, or in brazil at least. Unlike malaria, which I suppose it is routinely investigated in the areas of study, the authors must highlight the importance attached to the diagnosis of HPS.

In the methodology, the authors have chosen two sites of study according to their physiogeographical and population similarities but the gave no indication about these parameters ans clearly indicate the distance between the two areas. Thus, it would be pertinent to document the history of mouvement of the two cases of HPS.

In diagnosing the presence of Ig, did the authors use the same method (ELISA) for the two cases and the cohort ? Also did the use the same antigen(s) ? For the cohort, the used the N protein of Araraquara virus, is this antigen representative for all genotypes isolated in Brazil?

The authors gave a focus on socio-demographic parameters but they did not try to link it to the occurrence of HPS, otherwise is their a risk factor, especially related to housing condition, education...?

The prevalence of malaria is nearly the same as HPS prevalence, the authors must aknowledge the the technique used to diagnose malaria, despite that it is the gold standard, may not detect all malaria cases. Either, they should emphasize that malaria and hps may have the same level of circulation in their study area.

Author Response

In this study, the authors describe the occurrence of HPS and malaria in the particular context of gold mining.This stud is of interest as it is highlighting the difficulty to diagnose the etiology of febrile syndrome in tropical area and particularly in poor settings.The results of this study reinforce the the idea that that it is necessary to carry our a diagnosis of malaria before administering anti malarial treatment.

The study design is very clear and the results are well presented

However, the manuscript could be improved. In the introduction section the authors gave in a concise manner the prevalence of malaria and HPS in brasil by citing references, it would be nice to let know how these diseases, particularly HPS is occurring in the context of south America, or in brazil at least. Unlike malaria, which I suppose it is routinely investigated in the areas of study, the authors must highlight the importance attached to the diagnosis of HPS.

Authorsresponse: The corrections were inserted in the manuscript page 2 lines 68-76.

In the methodology, the authors have chosen two sites of study according to their physiogeographical and population similarities but the gave no indication about these parameters ans clearly indicate the distance between the two areas. Thus, it would be pertinent to document the history of mouvement of the two cases of HPS.

Authorsresponse: The corrections were inserted in the manuscript page 4 lines 139-141.

In diagnosing the presence of Ig, did the authors use the same method (ELISA) for the two cases and the cohort ? Also did the use the same antigen(s) ? For the cohort, the used the N protein of Araraquara virus, is this antigen representative for all genotypes isolated in Brazil?

Authorsresponse: The corrections were inserted in the manuscript page 4 lines 148-151.

The authors gave a focus on socio-demographic parameters but they did not try to link it to the occurrence of HPS, otherwise is their a risk factor, especially related to housing condition, education...?

Authors’response: The corrections were inserted in the manuscript page 5 lines 171-172.

The prevalence of malaria is nearly the same as HPS prevalence, the authors must aknowledge the the technique used to diagnose malaria, despite that it is the gold standard, may not detect all malaria cases. Either, they should emphasize that malaria and hps may have the same level of circulation in their study area.

Authorsresponse: The corrections were inserted in the manuscript page 7 lines 293-294.

Reviewer 2 Report

Dear Authors,

Thank you very much for the opportunity to read your manuscript. Your study is a good example of the importance of setting up appropriate differential diagnoses in areas with multiple infectious diseases with similar symptoms, such as in the Amazon region of Brazil. While I very much liked your manuscript, I do have several questions and comments below:

1.       The third and fourth sentence in the Abstract seems to be redundant. They both mention the two fatal cases of HPS, while the third sentence mention the seroprevalence study, and the fourth sentence mentions the importance of malaria. I suggest these two sentences to be combined to reduce the redundancy. Also, please first introduce a term (such as hantavirus pulmonary syndrome) before using the abbreviation for it (HPS).

2.       On page 2, line 59, you mention the term “legal Amazon”. What do you mean by that? Does that mean within the boundaries of the official Amazon region? Please clarify!

3.       The areas where the HPS cases occurred in the Uniao do Norte district seems like at least 600 km away from the Tres Fronteiras district in Coliza, where the seroprevalence survey occurred. That seems a long distance to me. On page 4, line 111-113, you state that there are physiogeographical and population similarities between these two areas. Could you please detail these similarities? Obviously, it would have been best to link the case study of these two cases with seroprevalence studies from the same population. However, I understand that that is not always possible, and having the opportunity of linking serum samples even from similar populations is still a unique opportunity. However, I do need to be more convinced that these two populations are at least comparable.

4.       On page 4, line 121-122, you state that the malaria incidence in Coliza was 4.5%, but the hantavirus seroprevalence was 3.57%. However, it looks like you had 4 malaria cases out of 112 study participants, and you also had 4 reactive IgG samples. Four out of 112 is 3.57%. So where did the malaria incidence of 4.5% come from?

5.       On Table 1, you list the socio-demographic characteristics of 112 residents in Coliza. Are any of the characteristics listed statistically significantly different between the anti-hantavirus IgG+ patients and the other participants? Alternatively, is the available data not sufficient for statistical comparisons? Perhaps methods developed for low sample size such as a Fisher’s exact would be helpful.

6.       How were the participants to this seroprevalence survey chosen? Are they an unbiased, representative sample of the overall population in this area?

7.       On page 5, line 135, you mention the term “vegetal exploration” (also in Table 1) as one of the occupations in the area. What does that mean? Is vegetal the same as “vegetable”? Is this than another agricultural activity? Please clarify!

8.       On page 5, Table 2, one of the options for Thick Blood Smear is “Positive for P. falciparum + FG”. What is FG? Please clarify!

9.       The Discussion at the bottom of page 5, and the top of page 6 starts with a description of the results of the epidemiological investigation related to the potential exposure of the case patients to hantavirus. This might be better placed at the case reports so that the reader can get the full picture there.

10.   On page 6, line 165, you list the word “precipitons”. What does that mean? Please clarify!

11.   On page 6, lines 180-188 provides a good description of hantaviruses in South America. This would be great in the Introduction. Please move it there!

12.   On page 6, line 194, you use the phrase “installation of new species”. Do you mean the invasion of new species? Or the establishment of new species?

Author Response

Thank you very much for the opportunity to read your manuscript. Your study is a good example of the importance of setting up appropriate differential diagnoses in areas with multiple infectious diseases with similar symptoms, such as in the Amazon region of Brazil. While I very much liked your manuscript, I do have several questions and comments below:

1.      The third and fourth sentence in the Abstract seems to be redundant. They both mention the two fatal cases of HPS, while the third sentence mention the seroprevalence study, and the fourth sentence mentions the importance of malaria. I suggest these two sentences to be combined to reduce the redundancy. Also, please first introduce a term (such as hantavirus pulmonary syndrome) before using the abbreviation for it (HPS).

Authors’response: The corrections were inserted in the manuscript page 1 lines 19-23.

2.      On page 2, line 59, you mention the term “legal Amazon”. What do you mean by that? Does that mean within the boundaries of the official Amazon region? Please clarify!

Authors’response: The corrections were inserted in the manuscript page 2 line 63.

3.      The areas where the HPS cases occurred in the Uniao do Norte district seems like at least 600 km away from the Tres Fronteiras district in Coliza, where the seroprevalence survey occurred. That seems a long distance to me. On page 4, line 111-113, you state that there are physiogeographical and population similarities between these two areas. Could you please detail these similarities? Obviously, it would have been best to link the case study of these two cases with seroprevalence studies from the same population. However, I understand that that is not always possible, and having the opportunity of linking serum samples even from similar populations is still a unique opportunity. However, I do need to be more convinced that these two populations are at least comparable.

Authors’response: The corrections were inserted in the manuscript page 4 lines 139-141.

4.      On page 4, line 121-122, you state that the malaria incidence in Coliza was 4.5%, but the hantavirus seroprevalence was 3.57%. However, it looks like you had 4 malaria cases out of 112 study participants, and you also had 4 reactive IgG samples. Four out of 112 is 3.57%. So where did the malaria incidence of 4.5% come from?

Authors’response: The corrections were inserted in the manuscript page 4 lines 152-153.

5.      On Table 1, you list the socio-demographic characteristics of 112 residents in Coliza. Are any of the characteristics listed statistically significantly different between the anti-hantavirus IgG+ patients and the other participants? Alternatively, is the available data not sufficient for statistical comparisons? Perhaps methods developed for low sample size such as a Fisher’s exact would be helpful.

Authors’response: The corrections were inserted in the manuscript page 5 lines 171-172.

6.      How were the participants to this seroprevalence survey chosen? Are they an unbiased, representative sample of the overall population in this area?

Authors’response: The corrections were inserted in the manuscript page 2 lines 142-144.

7.      On page 5, line 135, you mention the term “vegetal exploration” (also in Table 1) as one of the occupations in the area. What does that mean? Is vegetal the same as “vegetable”? Is this than another agricultural activity? Please clarify!

Authors’response: The corrections were inserted in the manuscript page 5 lines 174-175.

8.      On page 5, Table 2, one of the options for Thick Blood Smear is “Positive for P. falciparum + FG”. What is FG? Please clarify!

Authors’response: The corrections were inserted in the manuscript page 5 lines 182-185.

9.      The Discussion at the bottom of page 5, and the top of page 6 starts with a description of the results of the epidemiological investigation related to the potential exposure of the case patients to hantavirus. This might be better placed at the case reports so that the reader can get the full picture there.

Authors’response: The corrections were inserted in the manuscript page 3 lines 96-104.

10.    On page 6, line 165, you list the word “precipitons”. What does that mean? Please clarify!

Authors’response: The corrections were inserted in the manuscript page 6 line 199.

11.    On page 6, lines 180-188 provides a good description of hantaviruses in South America. This would be great in the Introduction. Please move it there!

Authors’response: The corrections were inserted in the manuscript page 2 lines 68-76.

12.    On page 6, line 194, you use the phrase “installation of new species”. Do you mean the invasion of new species? Or the establishment of new species?

Authors’response: The corrections were inserted in the manuscript page 6 line 219.

Reviewer 3 Report

General assessment and comments:

In the submitted manuscript, Terças-Trettel et al. reported the occurrence of hantavirus infection in gold-mining regions of Brazil and emphasized the importance of including hantavirus pulmonary syndrome (HPS) in differential diagnosis and performing hantavirus screen in the patents who living in the malaria-endemic area. The authors first presented two case reports of HPS that were confirmed by serological and molecular analysis. In addition, the authors described the result of a study that estimated the prevalence of anti-hantavirus antibodies in the mining area and showed evidence of co-infection of Malaria and hantavirus. 

This study addresses the importance of considering HPS in differential diagnosis of febrile infection in gold-mining regions of Brazil, which will provide helpful information to the readers in the field who are interested in this topic. Overall, the manuscript is well written. Here are some suggestions for authors to consider. 

(1) Line 74, I would suggest authors change “(Figure 1), associated with a study” to “(Figure 1) and a study”. 

(2) Line 107, change “Considering” to ”After”.

(3) line 120 change “enzyme immunoassays (ELISA)” to “enzyme immunoassays ELISA”

Author Response

General assessment and comments:

In the submitted manuscript, Terças-Trettel et al. reported the occurrence of hantavirus infection in gold-mining regions of Brazil and emphasized the importance of including hantavirus pulmonary syndrome (HPS) in differential diagnosis and performing hantavirus screen in the patents who living in the malaria-endemic area. The authors first presented two case reports of HPS that were confirmed by serological and molecular analysis. In addition, the authors described the result of a study that estimated the prevalence of anti-hantavirus antibodies in the mining area and showed evidence of co-infection of Malaria and hantavirus. 

This study addresses the importance of considering HPS in differential diagnosis of febrile infection in gold-mining regions of Brazil, which will provide helpful information to the readers in the field who are interested in this topic. Overall, the manuscript is well written. Here are some suggestions for authors to consider. 

(1) Line 74, I would suggest authors change “(Figure 1), associated with a study” to “(Figure 1) and a study”.

Authors’response: The corrections were inserted in the manuscript page 2 line 89.

(2) Line 107, change “Considering” to ”After”.

Authors’response: The corrections were inserted in the manuscript page 4 lines 133.

(3) line 120 change “enzyme immunoassays (ELISA)” to “enzyme immunoassays ELISA”

Authors’response: The corrections were inserted in the manuscript page 3 line 150.